# Longitudinal Assessment of Plasma Syndecan-1 Predicts 60-Day Mortality in Patients with COVID-19

**DOI:** 10.3390/jcm12020552

**Published:** 2023-01-10

**Authors:** Quan Zhang, Zhan Ye, Antonia Bignotti, X. Long Zheng

**Affiliations:** 1Departments of Pathology and Laboratory Medicine, The University of Kansas Medical Center, Kansas City, KS 66160, USA; 2Institute of Reproductive and Developmental Sciences, The University of Kansas Medical Center, Kansas City, KS 66160, USA

**Keywords:** COVID-19, endothelial dysfunction, glycocalyx degradation, syndecan-1, mortality

## Abstract

Background: Endotheliopathy is a common pathologic finding in patients with acute and long COVID-19. It may be associated with disease severity and predispose patients to long-term complications. Plasma levels of a proteoglycan, syndecan-1, are found to be significantly elevated in patients with COVID-19, but its roles in assessing disease severity and predicting long-term outcome are not fully understood. Methods: A total of 124 consecutive hospitalized patients with SARS-CoV-2 infection were prospectively enrolled and blood samples were collected on admission (T1), 3–4 days following treatment (T2), and 1–2 days prior to discharge or death (T3). Plasma levels of syndecan-1 were determined using an immunosorbent assay; various statistical analyses were performed to determine the association between plasma syndecan-1 levels and disease severity or the 60-day mortality rate. Results: Compared with those in the healthy controls, plasma levels of syndecan-1 in patients with critical COVID-19 were significantly higher (*p* < 0.0001). However, there was no statistically significant difference among patients with different disease severity (*p* > 0.05), resulting from large individual variability. Longitudinal analysis demonstrated that while the levels fluctuated during hospitalization in all patients, plasma syndecan-1 levels were persistently elevated from baseline in critical COVID-19 patients. Cox proportional hazard regression analyses revealed that elevated plasma levels of syndecan-1 (>260 ng/mL at T1, >1018 ng/mL at T2, and >461 ng/mL at T3) were significantly associated with the 60-day mortality rate. Conclusions: Endotheliopathy, marked by glycocalyx degradation and elevated plasma syndecan-1, occurs in nearly all hospitalized patients with SARS-CoV-2 infection; elevated plasma syndecan-1 is associated with increased mortality in COVID-19 patients.

## 1. Introduction

Endothelial dysfunction is a hallmark of systemic disease resulting from SARS-CoV-2 infection [1]. Endothelium is a dynamic cell layer involved in a multitude of physiologic functions, including control of vasomotor tone, trafficking of cells and nutrients, maintenance of blood fluidity, and growth of new blood vessels [2]. Once the endothelial layer is disrupted following infection [3], released toxin [4], trauma [5], irradiation [6], ischemia [7], etc., an endotheliopathy develops. Endotheliopathy is marked by an increased level of plasma von Willebrand factor (VWF), E-selectin, plasminogen activator inhibitor-1 (PAI-1), soluble thrombomodulin (sTM), and syndecan-1 [8], which is strongly associated with a poor outcome in patients with severe COVID-19 [9,10].

Syndecan-1 is a core backbone of vascular glycocalyx proteoglycans [11] that play an important role in regulating hemostasis, inflammation, trans-capillary flux, and microvascular permeability [12,13,14]. Plasma or serum levels of syndecan-1 on admission were found to be significantly elevated in patients with severe and critical COVID-19, which appears to be associated with mortality, intensive care unit (ICU) admission, and multiple organ damage [11,15,16,17,18,19]. However, the data published to date remain inconclusive owing to small sample sizes, and in most cases, a single time measurement. Thus, the role of assessing plasma syndecan-1 in hospitalized patients with SARS-CoV-2 infection in classifying disease severity and predicting outcome is still not clear. Here, we report a longitudinal assessment of plasma levels of syndecan-1 in these patients and its association with disease severity and long-term outcomes.

## 2. Methods

### 2.1. Patients and Sample Collection

The Institutional Review Boards of the University of Kansas Medical Center (KUMC) approved the study (IRB #00148313). The study was conducted according to the principles of the Declaration of Helsinki. One hundred and twenty-four consecutive hospitalized patients between January and March 2022 at the University of Kansas Hospital with a polymerase chain reaction (PCR)-confirmed SARS-CoV-2 infection and 20 healthy individuals were prospectively enrolled into the study. Venous blood was collected on admission (T1), 3–4 days following treatment (T2), and 1–2 days prior to discharge or death (T3) and anticoagulated with ethylenediaminetetraacetic acid (EDTA). Of 124 patients, 62 had more than 3 longitudinal blood samples and 72 patients had at least 2 samples collected. The exclusion criteria include those who were under 18 years old, patients who underwent a surgical procedure before the initial sample collection, or those with the initial sample collected 48 h after hospital admission. Patient demographic, clinical, and laboratory data were collected from electronic medical records at the time of blood collection. All patients were assessed for in-hospital adverse events and followed up to 60 days for obtaining all-cause survival information. Kidney Disease Improving Global Outcomes (KDIGO) criteria were used to define acute kidney injury (AKI).

### 2.2. Plasma from Healthy Controls

Age- and sex-matched healthy individuals who did not have acute and chronic inflammatory and thrombotic diseases or cancer were selected for the healthy controls.

### 2.3. Assay for Plasma Levels of Syndecan-1

Plasma syndecan-1 was determined using a commercially available enzyme-linked immunosorbent assay (ELISA) according to the manufacturer’s instructions (Abcam, Waltham, MA, USA). The intra- and inter-assay coefficients of variation are 6.2% and 10.2%, respectively.

### 2.4. Statistical Analysis

Continuous variables are expressed as the means ± standard deviations (SD) or the medians ± interquartile ranges (IQR) as appropriate. Mann–Whitney, Wilcoxon, or Kruskal–Wallis tests were used for paired and unpaired continuous variables, respectively. All categorical variables were described as numbers and percentages, and Fisher’s exact test (or χ^2^ test) was used for data analysis. Furthermore, Cox proportional hazard ratio regressions were employed to determine the most valuable predictor for 60-day mortality of COVID-19. To evaluate how variables can predict COVID-19 prognoses, the receiver operating characteristic (ROC) and Kaplan–Meier survival curves were calculated. All statistical analyses were performed using SPSS statistics version 26.0 (IBM, Armonk, NY, USA) and Prism 8.0 (GraphPad, San Diego, CA, USA).

## 3. Results

### 3.1. Demographic, Clinical, and Laboratory Characteristics of the Patients

A total of 124 consecutive hospitalized patients with PCR-positive SARS-CoV-2 were included in our study. These patients were categorized in asymptomatic, moderate, severe, and critical groups according to the updated World Health Organization guideline (https://www.mhcluster.org/WHO-2019-nCoV-therapeutics-2021.2-eng.pdf). Table 1 shows that there was no statistically significant difference in demographic features including gender, age, body mass index, and race among various groups. Additionally, a similar comorbidity rate was observed including diabetes mellitus, cardiovascular disease, chronic obstructive pulmonary disease, hyperlipoidemia, chronic kidney insufficiency, and history of malignancy and thrombotic events, except for hypertension, in which more patients exhibited hypertension in the critical group (*p* = 0.03). Of 124 patients, 14 (11.3%) were in the status of post-organ transplantation and immunosuppression, and 44 (35.4%) were admitted to intensive care unit (ICU), with the critical group having the highest rate of ICU admission (93.5%). There was a significantly higher incidence of sepsis or septic shock, AKI, and thrombotic events in patients with severe and critical COVID-19 than those with mild to moderate disease. Fourteen patients were intubated for mechanical ventilation and 15 died within 60 days following admission. Asymptomatic patients were those hospitalized for reasons other than COVID-19, including depression, coronary artery disease, alcoholism with alcohol withdrawal, hypothyroidism, trauma, post-operative infection, diabetic foot, burns, stroke, cirrhosis, end stage of renal diseases on hemodialysis, etc.

Table 2 summarizes the baseline laboratory data collected from COVID-19 patients within 24 h following admission. The results showed that the median levels of neutrophil proportion, lymphocyte count, serum protein, and D-dimers were significantly different in patients with severe and critical disease from those with asymptomatic and moderate disease. However, no statistically significant difference was detected in the baseline white blood cell count, neutrophil granulocyte and platelet counts, creatinine, or lactate dehydrogenase levels among all the groups.

### 3.2. Hospitalized Patients with SARS-CoV-2 Infection Exhibited Significantly Elevated Plasma Levels of Syndecan-1

Compared with the levels (median, IQR) in the healthy controls (49.5, 38.1–62.1 ng/mL), plasma syndecan-1 levels in hospitalized patients with SARS-CoV-2 infection on admission (156.8, 92.4–321.4 ng/mL) were significantly increased (*p* < 0.0001). To our surprise, statistically significant difference in plasma sydecan-1 levels (*p* < 0.0001) was only found in patients with moderate to critical disease compared with those in the healthy controls at all three time points (Figure 1A–C). These results suggest that endothelial damage appears to occur in all hospitalized patients with SARS-CoV-2 infection, regardless of their disease severity.

### 3.3. Longitudinal Changes of Plasma Syndecan-1 in Hospitalized Patients with SARS-CoV-2 Infection

Longitudinal blood samples were collected and assayed for plasma syndecan-1 levels in patients following SARS-CoV-2 infection during hospitalization. We found that in all hospitalized patients, their plasma levels (median 156.8, IQR 92.4–321.4 ng/mL) of syndecan-1 tended to increase initially (2–3 days) following treatments (median 191.7, IQR 124.2–567.0 ng/mL) (*p* = 0.003) (Figure 1D), then either slowly increased to reach a certain extent in those who survived (median 226.5, IQR 125.0–485.6 ng/mL) (*p* = 0.052) (Figure 1D,E), or persistently elevated in those who died (median 523.3, IQR 228.8–1074.0 ng/mL) (*p* = 0.007) (Figure 1D,F). These results suggest that endotheliopathy, indicated by an elevation of plasma syndecan-1, persists in patients with severe and critical COVID-19.

### 3.4. Elevated Plasma Levels of Syndecan-1 Are Associated with Mortality in Patients with COVID-19

To determine the clinical relevance of elevated plasma levels of syndecan-1, we performed the receiver operating characteristic curve (ROC) analysis, Cox proportional hazard risk assessment, and Kaplan–Meier survival analyses. ROC analysis demonstrated the cut-off value based on performance characteristic of sensitivity and 1-specificity using the cut-off. We found that plasma levels of syndecan-1 at T1 (>260 ng/mL) (*p* = 0.003) (Figure 2A), at T2 (>1018 ng/mL) (*p* = 0.03) (Figure 2B), and at T3 (>461 ng/mL) (p = 0.007) (Figure 2C) were associated with an increased 60-day mortality rate in patients with COVID-19. Cox proportional hazard risk analysis revealed that plasma syndecan-1 at T1 (>260 ng/mL) and T3 (>461 ng/mL) were strongly associated with the 60-day mortality rate in patients with COVID-19, with a hazard ratio (HR) of 7.1 (95% CI, 2.2–22.3) (*p* = 0.001) and 4.6 (95%CI, 1.3–15.9) (*p* = 0.015), respectively (Figure 2D). These results indicate that the plasma levels of syndecan-1 are predictive for an adverse outcome in hospitalized patients with SARS-CoV-2 infection.

## 4. Discussion

The present study demonstrates the role of longitudinal assessment of plasma levels of syndecan-1 in predicting 60-day mortality in patients with COVID-19. These novel findings are important but come to us with little surprise. The study addressed hard clinical endpoints including ICU admission, acute renal insufficiency (data not shown), and 60-day all-cause mortality rate. Only one patient died 26 days after discharge (40 days after admission). This patient was re-admitted after recovery from SARS-CoV-2 infection, although he died of end-stage lung and heart failure eventually. All other patients died of COVID-19 disease or COVID-19-related complications during hospitalization.

Syndecan-1 is released from the degradation of endothelial glycocalyx. It may be caused by direct infection of endothelial cells with SARS-CoV-2 following binding to outer membrane angiotensin-converting enzyme-2 (ACE-2) receptor [20,21]. It may also be the result of acute inflammation following viral infection [22]. Our correlation analysis provided evidence that an elevated plasma level of syndecan-1 was positively and moderately associated with inflammation biomarkers, including white blood cell count, C-reactive protein, D-dimer, and lactate dehydrogenase (data not shown). More recently, Lambadiari et al. [23] have demonstrated that when comparing with the healthy control, the thickness of glycocalyx in convalescent COVID-19 patients without hypertension and hypertensive patients decreases significantly, although the reduction of endothelial glycocalyx in these groups was not related to disease severity. These findings are consistent with ours, suggesting overlapping contributing factors to hypertension and SARS-CoV-2 infection.

Other potential mechanisms contributing to the elevation of plasma syndecan-1 levels include prolonged and overactive immune response with abnormality in cytotoxic T cells and monocytes [24]. Glycocalyx degradation may be mediated by reactive oxygen/nitrogen species, matrix metalloproteinases (MMPs), hyaluronidase, and deficiency of heparanase [25]. MMP-2, MMP-9, MT1-MMP, and ADAM-17 were shown to be capable of releasing syndecan-1 from endothelial surface [26]. Stahl et al. have demonstrated that, in patients with critical COVID-19, an acquired deficiency of heparanase-2 might contribute to an increased degradation of endothelial glycocalyx [27]. Overexpression of heparanase-2 or addition of heparin appears to prevent endothelial glycocalyx degradation in vitro [28,29,30], suggesting a therapeutic potential of targeting endothelial glycocalyx.

Elevated plasma syndecan-1 may not only serve as a biomarker for endothelial dysfunction and damage [31], but also contribute to a variety of pathophysiological functions [32]. Studies have shown that plasma syndecan-1 may contribute to inflammation [11,17] and thrombosis [33]. Plasma levels of syndecan-1 are associated with neutrophil activation and may reflect the degree of endothelial damage in patients with sepsis [34]. Syndecan-1 protein is found to be the major component of thrombi and plays a role of thrombus formation in a mouse model of anthrax [33]. In another animal study, shedding of proteoglycans may result in diminished thickness of the glycocalyx layer, leading to rapid adhesion and migration of leukocytes through injured endothelium [32].

Our study has some limitations. First, we did not have data from all follow-up samples due to early discharge, which makes comparison of longitudinal results less powerful; second, we did not recruit asymptomatic individuals who were not hospitalized as controls, or compare plasma syndecan-1 levels with other patients who had sepsis and ICU admission for other reasons than severe/critical COVID-19 disease, although such a comparison has been extensively performed in other previous studies and interpretation for the results would be complicated. Third, this is a single-center study, although our sample size is larger than those previously published in the biomarker studies [11,35]. Thus, a much larger and multi-center study with a longer follow-up should be conducted to confirm our findings. A more recent study demonstrates that prolonged endotheliopathy may be present even in patients with mid-to-moderate COVID-19 convalescence [36].

## 5. Conclusions

We conclude that plasma levels of syndecan-1 on admission and at other time points were significantly elevated in patients with severe to critical COVID-19 compared with the healthy controls. The elevated admission levels of plasma syndecan-1 may predict the all-cause 60-day mortality in patients with severe and critical COVID-19 disease. Our findings support the need for additional therapeutic strategies to improve endothelial health, thus reducing disease-related mortality in these patients.

## Figures and Tables

**Figure 1 jcm-12-00552-f001:**
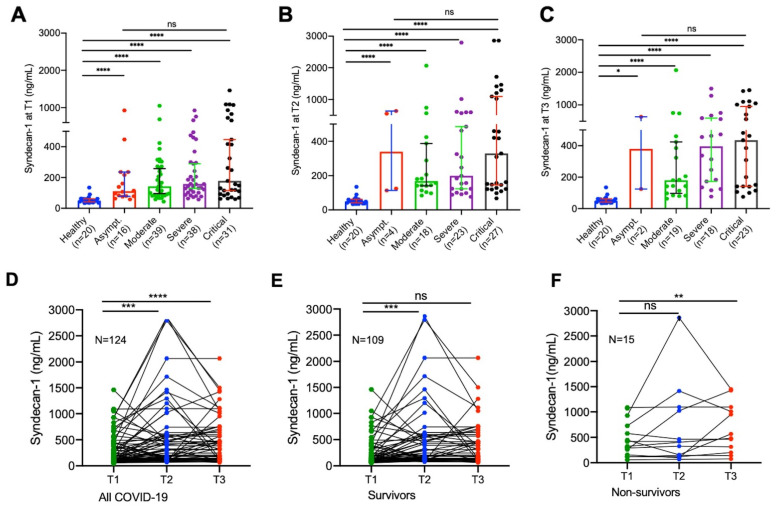
Plasma levels of syndecan-1 in hospitalized patients with SARS-CoV-2 infection and healthy controls. (**A**–**C**) Plasma levels of syndecan-1 in healthy controls (*n* = 20), asymptomatic patients, patients with moderate, severe, and critical COVID-19 at T1, T2, and T3, respectively. The data shown are individual values, the median, and interquartile range (IQR). Kruskal–Wallis test was performed to determine the statistical significance among the groups. Here, n.s. and **** indicate a *p* value >0.05 and <0.0001, respectively. (**D**–**F**) Longitudinal changes of plasma levels of syndecan-1 at T1, T2, and T3 in all patients with SARS-CoV-2 infection (*n* = 124), those who survived (*n* = 109), and those who died (*n* = 15) within 60 days of follow-up, respectively. Each individual value and trend at T1, T2, and T3 are shown. Paired *t*-test was performed to determine the statistical significance in plasma syndecan-1 levels of patients comparing each two groups. Here, n.s., *, **, ***, and **** indicate the *p* values of >0.05, <0.05, <0.01, <0.005, and <0.0001, respectively.

**Figure 2 jcm-12-00552-f002:**
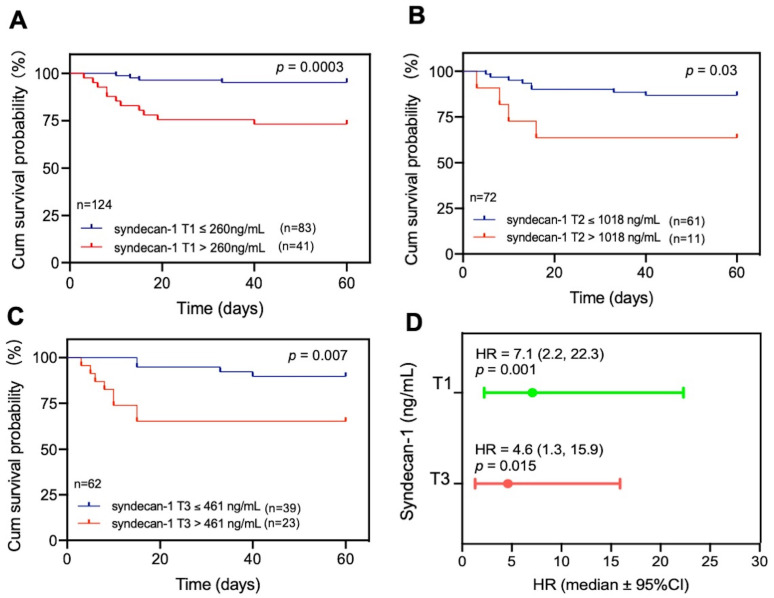
Elevated plasma levels of syndecan-1 in COVID-19 patients are associated with the 60-day mortality rate. Kaplan–Meier survival analysis determined the survival rates in COVID-19 patients with plasma levels of syndecan-1 >260 ng/mL vs. ≤260 ng/mL at T1 (**A**), >1018 ng/mL vs. ≤1018 ng/mL at T2 (**B**), and >461 ng/mL vs. ≤461 ng/mL at T3 (**C**). Cox hazard proportional regress analysis determined the hazard ratios (HR) of having increased levels of plasma syndecan-1 at >75 percentile at T1 and T3 (**D**).

**Table 1 jcm-12-00552-t001:** Demographics and clinical characteristics in 124 patients with SARS-CoV-2 infection.

Characteristics	Asymptomatic (*n* = 16)	Moderate (*n* = 39)	Severe (*n* = 38)	Critical (*n* = 31)	*p*-Value
Gender (male/female)	9/7	16/23	23/15	17/14	0.36
Age, year (mean ± SD)	56.2 ± 15.5	63.3 ± 16.5	63.9 ± 15.1	67.1 ± 13.3	0.15
BMI, kg/m^2^, median (IQR)	26.0 (24.4, 31.5) ^#^	29.1 (23.1, 33.5)	27.8 (23.4, 31.6)	28.9 (25.0, 32.3)	0.82
Races, *n* (%)					0.45
White	11 (68.8)	24 (61.5)	28 (73.7)	19 (61.3)	
Black	1 (6.2)	9 (23.1)	8 (21.1)	4 (12.9)	
Asian	1 (6.2)	2 (5.1)	0 (0.0)	2 (6.5)	
Others	3 (18.8)	4 (10.3)	2 (5.3)	6 (19.3)	
Comorbidities, *n* (%)					
Diabetes mellitus	2 (12.5)	9 (23.1)	8 (21.1)	14 (45.2)	0.06
Hypertension	2 (12.5)	19 (48.7)	18 (47.3)	18 (58.1)	0.03
CVD	2 (12.5)	8 (20.5)	8 (21.1)	11 (35.5)	0.31
COPD	0 (0.0)	5 (12.8)	6 (15.8)	5 (16.1)	0.40
Hyperlipidemia	1 (6.2)	12 (30.8)	8 (21.1)	12 (38.7)	0.08
CKI	1 (6.2)	7 (17.9)	5 (13.2)	3 (9.7)	0.69
History of malignancy	1 (6.2)	7 (17.9)	11 (28.9)	7 (22.6)	0.29
History of thrombotic events, *n* (%)	1 (6.2)	5 (12.8)	3 (7.9)	5 (16.1)	0.70
DVT	0 (0.0)	3 (7.7)	1 (2.6)	1 (3.2)	n.d.
PE	0 (0.0)	2 (5.1)	0 (0.0)	2 (6.4)	n.d.
DVT+PE	0 (0.0)	0 (0.0)	1 (2.6)	1 (3.2)	n.d.
Stroke	1 (6.3)	0 (0.0)	1 (2.6)	0 (0.0)	n.d.
MI	0 (0.0)	0 (0.0)	0 (0.0)	1 (3.2)	n.d.
Organ transplanted, *n* (%)	0 (0.0)	3 (7.7)	5 (13.2)	6 (19.3)	0.21
ICU admission, *n* (%)	2 (12.5)	3 (7.7)	10 (26.3)	29 (93.5)	0.00
Oxygen support, *n* (%)					
Nasal cannula	0 (0.0)	0 (0.0)	33 (86.8)	2 (6.5)	n.d.
Non-invasive ventilation	0 (0.0)	0 (0.0)	2 (5.3)	4 (12.9)	n.d.
High-flow oxygen	0 (0.0)	0 (0.0)	3 (7.9)	11 (35.5)	n.d.
Intubation	0 (0.0)	0 (0.0)	0 (0.0)	14 (45.2)	n.d.
Acute events, *n* (%)					
Sepsis or septic shock	0 (0.0)	0 (0.0)	1 (2.6)	13 (41.9)	0.00
AKI (KDIGO)	1 (6.2)	3 (7.7)	3 (7.9)	12 (38.7)	0.00
Thrombotic events	1(6.2)	0 (0.0)	6 (15.8)	7 (22.6)	0.007
Outcomes					
Length of hospitalization, day	3 (3, 7)	4 (3, 7)	7 (5, 10)	15 (10, 26)	0.00
60-day mortality, *n* (%)	0 (0.0)	0 (0.0)	3 (7.9)	12 (38.7)	0.00

Notes: ^#^ all data are presented as the median, interquartile range (IQR) unless specified otherwise. *p* values comparing four groups were obtained from the Chi-square (χ^2^) test, Fisher’s exact test, or Kruskal–Wallis test. *p*-values <0.05, <0.01 and <0.001 suggest statistically significant. SD—standard deviation; n.d.—not determined or not available; BMI—body mass index; COPD—chronic obstructive pulmonary disease; hyperlipidemia, defined by a total cholesterol >200 mg/dL or LDL > 100 mg/dL; CVD—cardiovascular disease; CKI—chronic renal insufficiency; DVT—deep vein thrombosis; PE—pulmonary embolism; MI—myocardial infarction; ICU—intensive care unit; AKI—acute kidney injury; and n.d.—not determined or not available.

**Table 2 jcm-12-00552-t002:** Laboratory parameters on admission in 124 patients with SARS-CoV-2 infection.

Parameters	Asymptomatic (*n* = 16)	Moderate (*n* = 39)	Severe (*n* = 38)	Critical (*n* = 31)	*p*-Value
WBC (×10^9^/L)	5.8 (4.5, 8.2) ^#^	7.1 (5.3, 9.5)	5.5 (3.9, 9.6)	7.9 (5.0, 11.8)	0.22
Neutrophil (%)	66 (57, 79)	72 (59, 79)	82 (68, 88)	88 (80, 93)	0.00
Lymphocyte (×10^9^/L)	1.3 (1.0, 1.5)	1.2 (0.9, 1.5)	0.5 (0.3, 0.9)	0.5 (0.3, 0.8)	0.00
Lymphocyte (%)	15 (9, 28)	15 (11, 23)	9 (6, 17)	5 (3, 10)	0.00
Platelet (×10^9^/L)	223 (173, 312)	230 (175, 282)	174 (122, 240)	221 (149, 311)	0.14
CRP (mg/dL)	n.d.	2.5 (1.1, 4.0)	8.1 (3.9, 17.6)	7.0 (4.4, 14.8)	0.08
D-Dimer (ng/mL FEU)	n.d.	1014 (394, 1377)	1544 (1014, 2999)	1932 (1088, 5620)	0.01
Albumin (g/dL)	4.3 (3.9, 4.6)	4.0 (3.4, 4.2)	3.3 (3.0, 3.7)	3.3 (2.9, 3.7)	0.00
LDH (U/L)	161 (135, 186)	211 (178, 376)	338 (209, 480)	342 (308, 413)	0.09
Creatinine (mg/dL)	0.8 (0.7, 1.1)	1.2 (0.9, 2.5)	1.2 (0.9, 2.4)	1.2 (0.7, 1.9)	0.13

Notes: ^#^ all data are presented as the median, interquartile range (IQR). WBC—white blood cells; CRP—C reactive protein; LDH—lactate dehydrogenase; and n.d.—not determined or not available.

## Data Availability

The datasets used and/or analyzed during the current study are available from the corresponding author at request.

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
