# Peer review of "Longitudinal Assessment of Plasma Syndecan-1 Predicts 60-Day Mortality in Patients with COVID-19"

_jcm, 2023, doi:10.3390/jcm12020552_

Round 1

Reviewer 1 Report

The work shows that 12% of the patients  studied unfortunalety die, however it is not specified if they die as a consequence of the previous comorbidity with wich they were admittedto the hospital or due to the consequences generated  during the hospitalization considered at 60 days according to their model (longitudinal-asseement) and if the causes of death are directly related to the high concentration os syndecan-1 and SARS-CoV2 virus or added to the other variables that favor endothelial dysfunction. So it is suggested  to discuss it at little more. It is recommended to cite the bibliographical references in a homogeneous way, since citation have the volume and other do not, as well as the first and last pages.

Author Response

The work shows that 12% of the patients studied unfortunately die, however it is not specified if they die as a consequence of the previous comorbidity with which they were admitted to the hospital or due to the consequences generated during the hospitalization considered at 60 days according to their model (longitudinal-assessment) and if the causes of death are directly related to the high concentration as syndecan-1 and SARS-CoV2 virus or added to the other variables that favor endothelial dysfunction. So, it is suggested to discuss it at little more.

We agree with the reviewer’s comments. In fact, the mortality we analyze is all-cause mortality. From our KM-curve (Figure 2A) we can figure out that only one patient died on 26 days after discharge (40 days after admission). This patient was re-admitted after recovery from SARS-Cov2 infection, though he died of end-stage lung and heart failure eventually (AECOPD + pulmonary hypertension). All other non-survivors died of COVID-19 or complications caused by COVID-19 during hospitalization. We have added a little more discussion in the text.

It is recommended to cite the bibliographical references in a homogeneous way, since citations have the volume and others do not, as well as the first and last pages.

Thanks for pointing out the inconsistency of citation formatting. Some of the references are ahead of print, so they do not have a volume so far. We have updated all the references as required in our tracked version.

Reviewer 2 Report

The present manuscript from Zhang Q. et al reports interesting data showing increased levels of sydecan-1 compared to healthy controls in all patients with COVID-19 regardless their severity.  Persistent levels of sydecan-1 are associated with increased mortality. Increased levels of sydecan-1 in patients with COVID-19 is consistent with previous study, however predicting long-term outcome was not fully understood. The manuscript is very well written and I have only a few minor comments.

1.     Could the authors provide some informations about healthy controls in patients and sample collection and in results.

2.     To bring out the increase or not of plasma levels of sydecan-1 could the authors indicate in the text the median value at T1, T2 and T3 (Fig. 1 D-E-F).

3.     In figure 2A use the same colour of figure 2 B-C (sydecan-1 > RED and sydecan BLUE).

4.     The Sydecan-1 in survivors patients seems to follow a bimodal distribution, some patients at T2 showed increased levels of Sydecan-1 while some patients showed decreased levels of Sydecan? There is some difference between these two groups?

5.     Sydecan-1 plasma levels were assessed in patients with COVID-19 and the authors found an elevation of plasma syndecan-1, that persists in severe and critical COVID-19.  The article is missing of an adeguate controls group compared to COVID-19 patients with similar comorbidities/ICU admission. Have the authors accounted for this in their conclusions?

Author Response

Could the authors provide some information about healthy controls in patients and sample collection and in results.

The information about healthy control group has been added in the Results section.

To bring out the increase or not of plasma levels of sydecan-1 could the authors indicate in the text the median value at T1, T2 and T3 (Fig. 1 D-E-F).

 The median values of plasma syndecan-1 at T1, T2 and T3 have been included in the revised text (please see tracked-changes).

In figure 2A use the same color of figure 2 B-C (sydecan-1 > RED and sydecan-1 ≤ BLUE).

Thanks for your comments, we have noticed that after submission and we have changed the curves to the same color in the revised figure 2A.

The Sydecan-1 in survivors patients seems to follow a bimodal distribution, some patients at T2 showed increased levels of Sydecan-1 while some patients showed decreased levels of Sydecan-1? There is some difference between these two groups?

This is a good way to analyze, but when we do the analysis, we find there are 39 (65%) increasing and 21 (35%) decreasing at T2 when compare to the level of syndecan-1 at T1. There is no significant difference in age, sex, BMI, and all other comorbidities between these two groups. The levels of syndecan-1 increase from 106 (78-181) ng/mL to 180 (127-605) ng/mL (p=0.000) in increase group, but decrease from 258 (148-494) ng/mL to 199.5 (124-424) ng/mL (p=0.014) in decrease group. The differences in baseline routine laboratory parameters we find are white blood cells (WBC) (decrease group 7.9,4.3-12.8 vs. increase group 5, 3.6-8.1 x109/L) (p=0.004), neutrophils (5.8, 3.2-9.5 vs. 4.4, 2.4-6.6 x109/L) (p=0.015), lymphocytes (0.68, 0.48-1.11 vs. 0.46, 0.3-0.92) (p=0.02), as well as level of syndecan-1 (p=0.000). But, there is no significant difference in level of syndecan-1 at T2 between the two groups (p=0.53). We don’t think it is necessary to add this to the text, which may cause some confusion. Also, the small sample size would not help make definitive conclusion.

Sydecan-1 plasma levels were assessed in patients with COVID-19 and the authors found an elevation of plasma syndecan-1, that persists in severe and critical COVID-19. The article is missing of an adequate controls group compared to COVID-19 patients with similar comorbidities/ICU admission. Have the authors accounted for this in their conclusions?

Thank you for the excellent comments and suggestions. However, because of the study design, we have already choses those SARS-Cov2 positive, but still asymptomatic patients who were admitted to hospital for other reasons than COVID-19 disease as the control group. We are sure that ICU patients with sepsis, or acute iTTP would for sure exhibit increase in endothelial damage marker such as syndecan-1. or comorbidities to be as another control will be perfect. However, this may complicate the interpretation of our data. In any case, our study has some limitation as we have discussed.

Round 2

Reviewer 1 Report

No comment.

Author Response

thank you.